# Neuropsychiatric Disorders and Frailty in Older Adults over the Spectrum of Cancer: A Narrative Review

**DOI:** 10.3390/cancers14010258

**Published:** 2022-01-05

**Authors:** Mariya Muzyka, Luca Tagliafico, Gianluca Serafini, Ilaria Baiardini, Fulvio Braido, Alessio Nencioni, Fiammetta Monacelli

**Affiliations:** 1IRCCS Ospedale Policlinico San Martino, 16132 Genoa, Italy; mariyamuzyka92@hotmail.com (M.M.); tagliaficoluca1992@gmail.com (L.T.); gianluca.serafini@unige.it (G.S.); ilaria.baiardini@libero.it (I.B.); fulvio.braido@unige.it (F.B.); alessio.nencioni@unige.it (A.N.); 2Department of Internal Medicine and Medical Specialties (DIMI), Section of Geriatrics, 16132 Genoa, Italy; 3Department of Neuroscience, Rehabilitation, Ophthalmology, Genetics, Maternal and Child Health (DINOGMI), Section of Psychiatry, University of Genoa, 16132 Genoa, Italy

**Keywords:** cancer, older adults, depression, anxiety, sleep disturbances, frailty, neuropsychiatry, attitude, motivation, support

## Abstract

**Simple Summary:**

Receiving a diagnosis of cancer in older adults supersizes the coexisting multimorbidity and frailty of the single individual. The multidimensional nature of this diagnosis needs to be appropriately targeted in the realm of the entire spectrum of cancer care, including the survivorship phase and the continuum of supportive care. The identification of late-life symptoms, syndromes, and the trajectory of frailty throughout the cancer course hold promise to better capture the clinical complexity of old-age patients and deliver targeted new pathways of care. In particular, neuropsychiatric disorders, beyond dementia, given their intrinsic association with frailty, need to be further explored to understand their impact on cancer disease course. Starting from this background, we aimed to assess the presence of neuropsychiatric conditions, including depression, sleep disturbances, anxiety, behavioral disturbances, attitude, motivation, and support in older adults receiving a diagnosis of cancer in order to understand the magnitude of the problem that may serve as a platform for future multidisciplinary studies and target interventions.

**Abstract:**

Background: The interplay between different neuropsychiatric conditions, beyond dementia, in the presence of a diagnosis of cancer in older adults may mediate patients’ fitness and cancer-related outcomes. Here, we aimed to investigate the presence of depression, sleep disturbances, anxiety, attitude, motivation, and support in older adults receiving a diagnosis of cancer and the dimension of frailty in order to understand the magnitude of the problem. Methods: This review provides an update of the state of the art based on references from searches of PubMed between 2000 and June 2021. Results: The evidence obtained underscored the tight association between frailty and unfavorable clinical outcomes in older adults with cancer. Given the intrinsic correlation of neuropsychiatric disorders with frailty in the realm of cancer survivorship, the evidence showed they might have a correlation with unfavorable clinical outcomes, late-life geriatric syndromes and higher degree of frailty. Conclusions: The identification of common vulnerabilities among neuropsychiatric disorders, frailty, and cancer may hold promise to unmask similar shared pathways, potentially intercepting targeted new interventions over the spectrum of cancer with the delivery of better pathways of care for older adults with cancer.

## 1. Introduction: Cancer in Older Adults

Aging is associated with an increased risk of receiving a cancer diagnosis. Although cancer remains a leading cause of death globally, the rapid development of new advances in diagnosis and treatments in recent years has paralleled with a substantial decline of mortality rates in most cancers, including those in older adults.

So far, the combination of these two factors has resulted in a growing percentage of cancer survivors in old-age individuals [1]. Cancer survival may be defined as the time lapse from the diagnosis of cancer to the end of life, and it brings new clinical challenges when facing the clinical complexity of older adults. In fact, biological aging is commonly associated with functional decline, multimorbidity, and frailty, a common geriatric syndrome characterized by reduced physiological reserve and ability to tolerate environmental stressors with a higher risk of health-related unfavorable outcomes, resulting in increased morbidity, disability, and mortality, and poorer quality of life [2,3,4].

Mounting evidence indicates that frailty is associated with cancer in older age and that frailty may affect a patient’s ability to tolerate cancer-related treatments and overall effectiveness [5,6]. Over half of older adults with cancer are estimated to have some degree of frailty [5,7,8] at the time of receiving a cancer diagnosis compared to the estimated prevalence of frailty that is of 12% measuring physical frailty and of 24% measuring frailty on the basis of the deficit accumulation model. Additionally, the pooling age group analysis of frailty in elders estimates 16% in the young old age group (60–69), 20% in the old age group (70–79 years), and 31% in the oldest old persons over 80 years [9], addressing the heterogeneity of frailty as non-liner function of age.

In addition, old-age cancer survivors are more likely to experience cognitive deficits and neuropsychiatric disturbances that may be partially the result of cancer and its related therapies. Hence, Lange et al. underscored that chemotherapy and other cancer treatments such as surgery, analgesia, hormones therapies, radiotherapy combined with immunotherapy, and targeted therapies could contribute to the development of neurocognitive deficits, expanding the spectrum of chemotherapy-induced neurocognitive impairment (CICI) [10].

So far, understanding the prevalence of neuropsychiatric disorders and psychological conditions in older patients with cancer has been largely unexplored, and questions regarding whether frailty and neurocognitive disorders may affect cancer-related outcomes are, in fact, unsolved. Similarly, the extent to which cancer and cancer treatment may shape frailty trajectories are unaddressed.

Starting from this background, here, we aimed to assess the presence of neuropsychiatric disorders, including depression, anxiety, sleep disturbances, along with psychological conditions such as attitude, motivation, and psychosocial dimensions like social support in older adults receiving a diagnosis of cancer.

Similarly, the dimension of frailty was assessed to understand the prevalence of these geriatric syndromes that may serve as a platform for future studies aimed at investigating their mediator role on old age cancer patients’ clinical outcomes.

## 2. Search Strategy and Selection Criteria

### 2.1. Data Sources

This review was based on a search of the MEDLINE database for articles in English between 2000 and June 2021 regarding the presence of neuropsychiatric conditions such as depression, sleep disturbances, anxiety, attitude, motivation, and support at the time of the diagnosis of cancer and the presence of frailty in older adults aged > 65 years.

The SANRA quality assessment for narrative review was used to assess the quality of data source [11]. Briefly, the six items that form the scale are rated from 0 (low standard) to 2 (high standard), with 1 as an intermediate score. The maximal sum score is 12. The sum score of the scale is intended to measure the construct “quality of a narrative review article” and covers the following topics: explanation of the review’s importance (item 1) and statement of the aims (item 2) of the review, description of the literature search (item 3), referencing (item 4), scientific reasoning (item 5), and presentation of relevant and appropriate endpoint data (item 6).

The definition and search for older adults were based on a chronological age of >65 years.

PubMedOvid Medline

### 2.2. Search Terms

Cancer (i.e., any type of solid and/or haematological cancer and any stage).Older adults OR old age OR aged (i.e., older adults > 65 years old). In geriatrics, the categorization of aging is based on the following stratification: young old (aged 65–74 years); old persons (75–84 years), and oldest old persons (>85 years).Neuropsychiatric disorders OR depression OR anxiety OR sleep disturbances OR attitude OR motivation OR support.Frailty definition and measurement on the basis of the conceptual framework of the physical frailty phenotype of Linda Fried [12] and/or the deficit accumulation model of Kenneth Rockwood [13] were used to select older adults with cancer.

### 2.3. Study Eligibility Criteria

We chose studies that included older adults >65 years of age, both sexes, receiving a diagnosis of cancer. Moreover, we selected studies that focused on the progression of cancer, including any cancer-related therapies (i.e., neoadjuvant, adjuvant, first- and/or second-line chemotherapy; radiotherapy; hormone therapies) and/or in cancer survivorship and/or in supportive care in the course of cancer and/or in late-life symptoms and syndromes.

We chose studies focusing on the development of neuropsychiatric conditions such as depression, anxiety, sleep disturbances, psychological conditions such as attitude, motivation, and the psychosocial dimension of social support at the diagnosis of cancer and/or in cancer survivorship and/or in supportive care in the course of cancer and/or in late-life symptoms and geriatric syndromes.

We selected studies focusing on the identification of frailty as a predictor of clinical outcomes based on the following clinical stratification:Short-term clinical outcomes (i.e., postoperative complications, 30-day mortality; length of stay (LOS) and 30-day readmission and/or cancer-related treatments’ toxicity and/or treatment non-completion and utilization of healthcare services).Long-term clinical outcomes (long-term mortality ≥1 year) and/or health-related and self-perceived quality of life along with long-term health-related quality of life (HRQOL) outcomes and/or reduced physical function.Late-life symptoms and/or late-life geriatric syndromes and/or identification or progression of late-life frailty.

Such studies were retrospective, prospective cohorts, observational or interventional in nature, with at least 50 included patients. Community-dwelling older adults with cancer and/or hospitalized patients were included in the study.

Exclusion criteria:AbstractsEditorials, case studies, score creation studies, pilot studies, and studies with fewer than 50 patientsStudies without a specific focus on older adults (i.e., age < 65 years or no data about old-age participants)Articles related to central nervous system cancer, childhood and adult cancersStudies on specific single-disease-affected populationsNursing home patientsOlder adults comorbid for neurocognitive conditions, such as all types of dementia (Alzheimer’s dementia, vascular dementia, rapid progressing dementia, Parkinson’s dementia, Lewy body dementia, frontotemporal dementia) according to the current diagnostic criteria and guidelines, and/or medical record and clinical history and/ORCDR scoring) and/or severe depression (as assessed with any validated scale for depression and/or by physician’s clinical judgment and/or DSMV criteria), prior to the diagnosis of cancerOlder adults with moderate to severe multimorbidity (as assessed by cumulative illness rating scale > 6 or a number of co-occurrent diseases > than 6)Palliative cancer patientsEnd-stage single-disease patients with cancer such as advanced renal failure, advanced cardiac failure, and advanced lung disease

Figure 1 illustrated the selection process. Overall, 11,286 patients as overall sample were analyzed on the basis of the finally included studies.

## 3. Frailty

### 3.1. Definition of Frailty

Frailty in aging marks a reduced physiologic reserve with increased vulnerability when exposed to environmental stressors. Although the concept of frailty in aging seems to bring a somewhat qualitative concept, a meaningful association between frailty and higher disability and mortality is growingly observed in old-age cancer patients [4].

To a similar extent, cancer and its treatment are associated with acceleration in the progression of frailty [14] because surgery, the perioperative period, and chemotherapy may be considered specific environmental stressors that affect the individual functional reserve. Therefore, Margolick and Ferrucci underscored that cancer may be claimed as a potential accelerator of biological aging because cancers’ driven anatomic and functional manifestations, their underlying mechanisms, and the impact of cancer therapies could also be caused by aging, although detected at a younger age than usual [15].

So far, the standardization of the frailty construct in clinical practice has been highly heterogeneous, with a redundant abundance of frailty definitions and clinical measurements. To overcome this heterogeneity, two main conceptual frameworks for frailty—the physical frailty phenotype of Linda Fried (FP) [12] and the deficit accumulation model of Kenneth Rockwood (FI) [13]—were developed.

Namely, these two distinct conceptualizations bring similar predictions for high mortality and institutionalization risk, but they denote differences in measurement and process, potentially identifying different populations and interventions. Namely, the physical frailty phenotype of Linda Fried (FP) is a state distinct from multimorbidity and disability, resulting from declines in multiple physiological systems, particularly in metabolic and musculoskeletal systems, with underlying specific biological culprits. The physical frailty phenotype is the clinical presentation of the syndrome and is characterized by the presence of three out of five of the following criteria: weakness, slowness, low physical activity, exhaustion (or fatigue), and unintentional weight loss (0, robust; 1–2, prefrail; ≥3, frail) [14].

The deficit accumulation model of Rockwood (FI) is based on a large set of clinical conditions that create an individual aggregated risk of poorer outcomes. Its operationalization is in a frailty index wherein the deficits can include symptoms, signs, medical conditions, polypharmacy, cognitive impairment, functional impairment and poor mobility, balance, and laboratory biomarkers. This FI is calculated as the proportion of health deficits present in a given individual [15].

Notwithstanding the adopted frailty conceptual framework, both the International Society of Geriatric Oncology and the American Society of Medical Oncology underscored that all older adults with cancer should undergo an early frailty stratification to accurately balance the harm-to-benefit risk for therapeutic aggressiveness and prognosis, tailoring individualized interventions [16,17].

### 3.2. Frailty and Cancer Outcomes

In the realm of oncogeriatrics, the majority of studies include the measurement of the physical frailty phenotype of Fried. Namely, Pamukcuoglu et al. performed a prospective longitudinal study of haematopoietic cell transplant (HCT) recipients and frailty, assessed before HCT, to predict severe non-haematological toxicities, non-neutropenic infections, and pneumonia. Frail patients also had a 3.1 times higher risk of overall mortality as compared with non-frail allogeneic HCT recipients [18].

In the field of haematology, Murillo et al. conducted a study comparing prospectivefrailty according to the physical frailty phenotype with the International Myeloma Working Group (IMWG) scoring system, which includes age, deficits in activities of daily living (ADLs), impairments in instrumental ADLs (IADLs), and the Charlson comorbidity index (CCI) [19]. Of 98 patients with a median age of 79 years, the frailty category when using the Fried model was significantly associated with a higher risk of mortality.

In surgical oncology, Tan et al. studied a population of 83 patients undergoing elective colorectal cancer resection [20]; those with frailty syndrome were found to have a 4-fold higher risk of developing major complications of both a surgical and medical nature.

Similarly, Kristjansson et al. compared frailty on the basis of both the physical phenotype of Fried and the comprehensive geriatric assessment (CGA) in a cohort of 176 older adults electively undergoing colorectal cancer, showing the good predictive value for both assessments in predicting overall one-year survival [21].

The physical frailty phenotype was also used in bladder cancer older adults. Namely, 123 patients undergoing radical cystectomy (RC) with a median age of 74 years were assessed prospectively for frailty to predict postoperative complications. Being intermediately frail or frail was associated with high-grade 30-day postoperative complications [22]. Interestingly, Feliciano et al. showed in a multicenter, prospective cohort study including 7257 postmenopausal women [23] that a frailty diagnosis at baseline, on the basis of the physical frailty phenotype, was highly predictive of poorer 3-year overall survival.

Runzer-Colmenares et al. showed that frailty assessed by the physical phenotype of Fried was associated with the development of radiotoxicity in a cohort of 181 geriatric patients [24]. There was also a good correlation between vulnerability assessed by the Vulnerable Elders Survey-13 (VES-13) or G-8 questionnaire screening tool and radiotoxicity. The same authors conducted another retrospective cohort study to evaluate the association between frailty with the development of chemotherapy toxicity in oncogeriatric patients [25] and showed that the physical phenotype of Fried showed the strongest association with mortality.

On the one hand, Hay et al. investigated the potential association between the frailty physical phenotype and chemotherapy tolerability [26] and underscored that frailty was associated with a higher rate of non-initiation of standard-of-care chemotherapy and with increased withdrawal to complete chemotherapy in adjuvant patients. On the other hand, Haddad et al., in their real-life population study, did not find an association between cancer and frailty severity, using both a phenotypic model and the deficit accumulation model [27]. The authors concluded that an overall accumulation frailty model over time and the role of age, gender, and comorbidity, including cancer stage and severity, might shape the clinical frailty trajectories, favoring poorer clinical outcomes.

So far, scant evidence has underscored the association between the deficit accumulation model of Rockwood and the main clinical outcomes in older adults with cancer. Inci et al. performed a prospective study enrolling 144 patients with ovarian cancer undergoing cytoreductive surgery, evaluating the predictive value of the frailty index on severe postoperative complications and overall survival [28]. The main results showed that a frailty index >0.26 (OR 3.64, 95% CI: 1.34–9.85, *p* = 0.01) was highly predictive of severe postoperative complications, whereas a frailty index >0.15 (HR 1.87, 95% CI: 1.01–3.47, *p* = 0.048) showed poor survival during a median follow-up observation of 37.6 months.

Additionally, Giannotti et al, in a cohort of 99 geriatric patients candidate for elective gastrointestinal cancer surgery, showed the non-inferiority accuracy of the 40-item frailty index (FI) compared to CGA for the prediction of both short-term mortality and 1-year mortality [29]. Table 1 illustrates the core studies, which includes the measurement of frailty and clinical outcomes in old-age cancer patients on the basis of both the physical frailty phenotype of Fried and/or the deficit accumulation model of Rockwood.

### 3.3. Supportive Studies with Respect to Core Studies

Several frailty-screening assessments as surrogates of the aforementioned main conceptual framework for frailty have been developed to assess and predict clinical outcomes in older patients with cancer. In surgery, the modified frailty index (mFI) was developed using 11 out of the 70 items in the Canadian Study of Health and Aging Frailty Index [30] and was found to be a reliable predictor of short-term complications, including 30-day mortality and major morbidity as far as post-surgical complications, longer length of stay (LOS) in hospital with increased hospital costs in gastrointestinal cancer surgery [31,32,33,34] as well as pancreatic [35], gynecologic [36,37], head and neck [38,39], and urologic [40,41] cancers.

Moreover, frailty in breast cancer patients was also associated with moderate and severe chemotherapy-related toxicity (grade 3–4), reduced the treatment complexion rate of adjuvant therapies, and, as a result, poorer treatment outcomes were observed [42,43,44,45].

As far as the hematological cancer setting is concerned, a recent review by Scheepers et al. demonstrated that frailty was associated with both short- and long-term (1 year) mortality [46]. In addition, a meaningful correlation with treatment-related toxicity (hematologic and overall toxicity) along with a higher risk for treatment non-completion was found. Principle scores used in this field are IMWG frailty (age, medical comorbidities (assessed by CCI) and disabilities (assessed by ADL score and IADL score)), associated with 3-year overall survival and progression-free survival [47]. In 2017, Engelhardt et al. developed a scoring system referred to as the revised myeloma comorbidity index (R-MCI) on the basis of age, Karnofsky performance status (KPS), estimated glomerular filtration rate (eGFR), frailty, cytogenetics and lung dysfunction [48]; this system showed good prognostic value for overall survival.

Some evidence recently underscored how a cancer diagnosis may have a negative impact on the quality of life. Namely, frailty was associated with worse health-related and self-perceived quality of life along with long-term health-related quality of life (HRQOL) outcomes [49,50]. In the first study, frailty was assessed with a modified 38-item TOPICS-MDS frailty index, which originally consisted of 46 items. The second one used a 36-item Carolina frailty index (CFI) based on the principles of deficit accumulation, including multiple items relating to limitations in IADL, comorbidities, cognition, social activity, falls, and nutrition.

### 3.4. Biomarkers of Frailty and Cancer

Growing biological evidence supports frailty as an aging construct. Indeed, measuring and grading the biomarkers of frailty is one approach for determining biological age and reflects the complexity and heterogeneity of aging despite chronological age [50,51,52,53,54]. 

Moreover, biological aging is associated with the process of immunosenescence and so-called “inflammaging”. Namely, inflammaging is a low-grade age-related systemic inflammation that interplays with immunosenescence, which, in turn, involves a gradual deterioration of the immune system related to thymic atrophy, reduced immunosurveillance, and a decrease of T-cell, B-cell, and dendritic cell function [51]. Both age-related processes are increasingly believed to play a key role in the development of frailty as well. Immunosenescence may favor aging and cancer progression by virtue of reduced immunosurveillance and accumulation of senescent cells that affect the microenvironment with functional changes in an aging immune system. Hence, the intertwined inflammaging network, related to the activation of inflammasomes and danger-associated molecular patterns (DAMPs), is boosted by the reduction of autophagy capacities, microbiota barrier permeability alterations, and changes in cellular chromatin, creating a unique pro-tumorigenic environment. In turn, inflammation may further trigger multiple stressors, including DNA damage, supporting multiple degenerative cancer-related processes [52].

In the context of immune system age-related alterations and on the basis of the two main constructs for frailty, a series of biomarkers has been associated with the development of frailty. In particular, interleukin 6 (IL-6), tumor necrosis factor-α (TNF-α), C-reactive protein (CRP), and white blood cells (WBCs) were observed as associated biomarkers in both the physical frailty phenotype and the deficit accumulation model of Rockwood [53]. Moreover, immune system biomarkers, such as T follicular helper cell (Tfh cell) subsets, interleukin-1 receptor antagonist(IL-1Ra), and soluble endothelial leukocyte adhesion molecule-1 (sE-selectin), were associated with the deficit accumulation model of Rockwood, whereas CD8+CD28−CD27+, dysregulation of CD4 T, C-X-C motif chemokine ligand 10 (CXCL10),and transforming growth factor- β (TGF-β) were associated with the physical frailty phenotype [51,54,55,56].

Regarding frailty, endocrinosenescence seems to play a key role, and hormone changes demonstrated the strong association with musculoskeletal alterations and sarcopenia that, as a result, drive clinical frailty. In particular, testosterone levels, dehydroepiandrosterone (DHEA), insulin-like growth factor-1 (IGF-1), hyperparathyroidism, adiponectin, leptin, and vitamin D deficiency were associated with both frailty constructs, whereas insulin-like growth factor binding protein 1–3 (IGFBP 1–3) was related to the deficit accumulation model of Rockwood [53,55,56]. So far, vitamin D deficiency has shown the strongest correlation with frailty on both theoretical constructs, driving the transition from fit through vulnerable and towards an overtly frailty phenotype [53].

In addition, haemoglobin, albumin, mitochondrial haplogroup and APOE genotype, a low glomerular filtration rate, and reduced telomere length were associated with both frailty constructs. Conversely, markers of, for example, advanced glycation end products, protein carbonyls, oxidized lipoproteins, and antioxidant deficiencies were associated with the physical frailty phenotype and altered DNA repair and DNA damage/DNA repair ratio was associated with the deficit accumulation model of Rockwood [53,56].

However, it could be hypothesized that the combination of different biomarkers might fill the gap of measuring such a complex biological aging process, and although a standard definition of frailty is missing, the identification of the two constructs for frailty may add knowledge in the field of biomarkers as well [57].

Tentatively, in Figure 2, the main intertwining association between frailty and biological aging on the basis of the moderating role of inflammaging, immunosenescence, and endocrinosenescence is illustrated. This pathogenetic background may serve for a further understanding about the interrelation between cancer and frailty in older adults as well. Namely, aging, frailty, and cancer are highly correlated phenomena and share several underlying mechanisms, including DNA damage responses and cellular senescence that might serve as boosters of aging phenotypes. On the other hand, aging can be considered a pro-tumorigenic state, conferring a higher risk for cancer development. Biomarkers of aging are prevalent in cancer survivors, and increasing research is aimed at identifying the biomarkers of frailty that could reflect the possible intertwined pathophysiology with cancer in old-age patients in order to provide targets for future interventions [58].The identification of biomarkers associated with frailty is preliminary and speculative in nature, and further investigations into the interplay between the markers of frailty and cancer is warranted to capture common pathophysiological mechanisms that may favor further advances in clinical practice and treatment for old-age patients with cancer [53,59,60,61,62,63,64,65,66,67,68,69,70,71,72,73].

### 3.5. Neuropsychiatric Disorders in Older Adults with Cancer

#### 3.5.1. Depressive Disorders and Suicidal Ideation in Older Adults with Cancer

Older adults receiving a diagnosis of cancer need to be considered at higher risk of developing depressive disorders and death from suicide. Based on a meta-analysis of interview-based studies, major affective disorders may be particularly common in older patients with cancer, with at least 30–40% of patients reporting that they had suffered from mood disturbances in hospital settings [74]. However, affective spectrum disorders are not only common in this vulnerable population but are even linked to unfavorable clinical outcomes in terms of poorer quality of life, reduced cancer-related treatment adherence, negative cancer illness trajectory, and higher subjective perception of physical symptoms. When older adults with cancer are diagnosed with depression, it is also imperative that clinicians assess them accurately to identify any potential signs of suicidality to rapidly and comprehensively manage the suicidal risk [75].

Notably, a significant burden of neuropsychiatric disorders, including higher depressive burden, may be documented even after 10 years from the initial diagnosis, especially in older males with localized prostate cancer [76], driving poorer late-life syndromes, increased functional decline, and frailty. Indeed, due to the intertwined nature of both frailty and depression, the early identification of depression and its effective treatment management should include tailored psychological interventions as supportive care in the spectrum of cancer and might hold promise in reducing the overall burden of frailty, improving late-life outcomes, and quality of care.

So far, late-life depression in older adults represents a prevalent clinical condition [77,78] that has been misdiagnosed to a significant degree and is tightly associated with higher disability and multimorbidity, driving unfavorable clinical outcomes [79,80,81].

In the realm of cancer, the evidence fails to capture the specificity of depressive phenotypes associated with cancer, and, as a result, no standardized construct for its definition, clinical assessment, and treatment is currently available. However, the early identification of loneliness, hopelessness, and demoralization may help clinicians to early identify older individuals at major risk of developing clinically relevant depression and suicidal behavior. In particular, the so-called “demoralization syndrome” may be defined as a new diagnostic entity occurring in hospitalized older patients with cancer in different stages of the disease spectrum that is associated with disability, functional decline [82], and has been recently proposed as a clinical precursor of major depression linked to increased suicide risk in this highly vulnerable population.

Similarly, the effectiveness of antidepressant medications and psychotherapy in older cancer patients with depression invoke a multidisciplinary assessment to tailor newly supported care strategies that could positively impact this highly prevalent old-age neuropsychiatric condition [83].

It is noteworthy that, even in the absence of any clinically relevant symptoms of depression, suicide risk cannot be disregarded in older cancer patients. According to population-based cancer registry studies, the incidence of suicide among older adults with cancer is nearly double relative to the general population [84,85,86,87,88,89], and after adjusted analyses, cancer maintained the strongest association with suicide [90].

Increased suicide rates have been reported within the first three months after a cancer diagnosis [91]. Based on standardized mortality ratios, in a US cohort study, patients with lung, bronchus, or stomach cancers were at least five and four times more likely to die by suicide, respectively, as compared to the general population [86]. Similarly, 8.6% of cancer patients in a Japanese old-age cohort manifested suicidal ideation, which is widely considered to be a reliable predictor of suicidal behavior [92].

Overall, suicide risk is of paramount importance for older adults with cancer, and its prevention needs to be addressed as a key priority. Given the intrinsic vulnerability of older patients with cancer and their higher risk for suicide, clinicians should consider the existence of specific risk factors in this population relative to individuals with cancer and depression. Thus, the time frame since a cancer diagnosis disclosure [91,93], the types of cancer with high fatality [94], and the advanced stage of illness [95] must be recognized as unique risk factors in older patients with cancer in combination with their pre-existing clinical vulnerability, multimorbidity, and frailty. Moreover, a positive history of prior suicide attempts, psychiatric disorders, psycho-social difficulties and lower social support, chronic pain, and a family history of suicide [96] may also be considered highly relevant risk factors for suicide ideation. This constellation of symptoms may assist physicians to draw a more comprehensive framework for the appropriate identification and the development of an appropriate clinical and therapeutic management, which is still warranted [97,98] to avoid unfavorable outcomes (e.g., suicidality). The systematic and careful assessment of suicide risk should be considered an absolute priority in older cancer patients with the need to rapidly and appropriately identify vulnerable at-risk populations in order to promote targeted and tailored interventions.

#### 3.5.2. Cancer and Anxiety Symptoms in Older Adults with Cancer

Anxiety and distress are common findings in older adults with cancer. The fear of death in older individuals is typically experienced as the result of disclosing a diagnosis of cancer [99,100]. In particular, the direct communication of the diagnosis may induce pervasive anxiety, disabling distress and intolerable stress levels [101,102]. Anxiety may be considered a highly frequent psychological problem, particularly for patients diagnosed with incurable cancer [90,103,104].

Death anxiety may be influenced in older subjects by the type of cancer, gender, and marital status [105]. Consistent evidence has reported that cancer patients manifest in at least 10–12% of cases comorbid clinical anxiety disorders, with subjects with advanced cancer representing a specific vulnerable subgroup of patients [74,106].

Notably, the highest levels of anxiety were observed in patients with haematological, gynecological, and lung cancers [107]. Although risk factors for the development of anxiety and depression are poorly understood, social deprivation, as well as female gender, were found to be relevant risk factors, especially for colorectal and lung cancer [108]. Moreover, pain and anxiety with the presence of at least one of these conditions worsened the subjective disabling experience of the other [109], and nearly two-thirds of cancer patients with depression also manifest clinically significant anxiety symptoms [110].

The invasive potential of some medical procedures in older patients with cancer and the need to be closely monitored may be a common psychological factor involved in the perceived distress and anxiety feelings in these patients; thus, the implementation of evidence-based interventions aiming to reduce the burden of these symptoms are of paramount importance [109,111]. However, different biological mechanisms have been proposed to explain the increased rates of depression and anxiety in patients with incurable diseases such as cancer. For instance, depression and anxiety may directly affect endocrine and immune functions [112], with the well-known dysregulation of the hypothalamic–pituitary–adrenal (HPA) axis seeming to be implicated as a possible mediator of vulnerability to the occurrence of hormone-related cancers. The abnormal activity of natural killer (NK) cells and DNA repair enzymes has also been proposed as being altered in depressed individuals [113]. Furthermore, there are maladaptive lifestyle habits and daily life conditions such as being sedentary or increased levels of alcohol use and smoking that may represent additional risk factors for anxiety and cancer [114,115].

Unfortunately, anxiety disorders are often undetected and undertreated in the clinical practice due to the overlap of anxiety, cancer symptoms, and adverse effects related to cancer treatment [116], particularly in older individuals when anxiety was linked to detrimental social interaction, cognitive and emotional functioning, poor nutrition, and comorbidity [117]. Supportive care is now strongly recommended for its effectiveness, particularly regarding quality of life, anxiety and depression management, and caregiver distress during the course of cancer [118,119]. Another clinically relevant issue is represented by the subgroup of cancer survivors who may face severe and invalidating further physical/psychosocial distress and additional symptoms in terms of fatigue, cognitive impairment, depression, anxiety, and stress levels [120,121]. Hence, the identification of the global burden of neuropsychiatric disorders related to being cancer survivors is fundamental for clinicians to provide better supportive care throughout the entire spectrum of cancer disease.

#### 3.5.3. Sleep Disorders in Older Patients with Cancer

So far, aging has primarily been associated with a reduced ability to maintain sleep, with sleep quantity and quality disruption occurring frequently, leading to potential health implications. Although with unclear prevalence rates [122], several studies reported the association between sleep disturbances and cancer, particularly in older individuals [123,124,125]. In existing studies limited to patients with cancer who were aged 65 and older, the prevalence of insomnia was reported to vary from 19 to 60% [126,127,128].

Sleep disorders have emerged as potential cancer risk factors, with insomnia, circadian disruption, obesity, and intermittent hypoxia in obstructive sleep apnea being reported as contributing risk factors for an increased risk of several types of cancers; in particular, stomach, esophageal squamous cell, and breast cancers have been reported in short sleepers [129]. Unfortunately, on the one hand, studies aimed at appropriately defining the nature of sleep disorders in patients with cancer may be influenced by the fact that participants might be undergoing radio/chemotherapy and may manifest sleep disturbances as a result of their treatments. On the other hand, a poor sleep-to-wake cycle is associated with impaired genotoxic stress counterbalance and DNA repair with boosted chemotherapy-related effects.

Moreover, sleep-wake disturbances are associated with incident dementia, functional decline and frailty. Moreover, higher psychological stress, such as in the presence of a cancer diagnosis, may trigger sleep disruption, favoring frailty trajectories as well. Therefore, sleep disturbances may themselves be considered to be a pathophysiological expression of altered circadian rhythms with persistent perturbation of homeostasis, leading to diminished functional reserve over time.

Additionally, a bidirectional relation has been proposed between sleep disorders and cancer, underpinning some common underlying molecular mechanism [130]. Various authors have hypothesized a dysfunction in thermoregulation, appetite control, motivation, and sleep that is directly related to hypothalamic abnormalities. In addition, an abnormal release of hypothalamic orexin has been postulated, with concomitant alterations regarding other neurotransmitters and/or neuropeptides. However, hyperactivity of the HPA axis, together with an abnormal production of glucocorticoids and autonomic dysfunctions with cortical activation and consequent impairment of circadian rhythms, have also been reported. Importantly, an increased activation of the inflammatory response, particularly interleukin-1 β (IL-1 β), which is implicated in the enhancement of microglia neurotoxic reaction, has been noted in subjects with sleep disorders, even during chemotherapy. IL-1 β is also able to inhibit REM sleep, promote non-REM sleep, influencing neurotransmitter (serotonin, dopamine, GABA, and noradrenaline) concentrations involved in the quality of sleep. Specific hormones, such as ghrelin and leptin, are similarly able to abnormally stimulate inflammation. Notably, the increase of additional mediators involved in enhanced hyperarousal state, such as catecholamine release and sleep fragmentation, have been reported as well [130].

According to recent evidence, the most relevant spectrum of sleep disorders associated with cancer in older patients includes insomnia, hypersomnia, and circadian rhythm disturbance [131]. Interestingly, sleep issues were reported to be independently associated with cancer, even after controlling for the presence of fatigue and depression [132]. Older adults with cancer reporting fatigue and sleep disorders often have coexisting geriatric syndromes (comorbidities, polypharmacy, dementia, delirium, depression, and/or falls) and are at high risk of incipient functional decline [125]. Sleep disturbances may be common even in cancer survivorship; indeed, some years after the discontinuation of radio/chemotherapy treatment, the persistence of a sleep disturbance is a key predictor of disability and psychosocial impairment [133]. Thus, the careful evaluation of sleep disorders in old-age patients with cancer is a key priority for improving their overall quality of life, potentially reducing their late-life frailty burden.

Overall, a close relationship between sleep disorders and cancer has been postulated, with the potential of novel treatments to identify sleep disturbances as possible targets of the response to chemotherapy and immunotherapy agents [130]

#### 3.5.4. Psychological Conditions: Attitude and Motivation

##### Attitude

Besides the presence of neuropsychiatric disorders, there are further psychological conditions and psychosocial dimensions that may be obstacles to the management of older adults with cancer patients across the disease course. Research has highlighted that attitude [134], motivation [135], and support [136,137] play a role in determining health outcomes, choices, and behaviors in a variety of cancer patient populations. They are potentially modifiable and may, thus, represent targets to improve decision-making processes and personalized care and treatment.

Currently, few studies have assessed these factors in the life context of older patients with cancer, analyzing specific aspects of subjective experience.

Attitude toward own aging (ATOA) perceptions, cognitive representations, and expectations of individuals regarding their own aging have been found to affect the physical health of individuals with chronic illnesses through a stress-related pathophysiological pathway [138,139]. A recent study by Martin et al. [140] extended this research by exploring ATOA in older adults (219 with cancer and 912 without cancer). The results indicated that ATOA is associated with physical and mental health, but not cognitive function, to a similar extent across individuals with and without cancer. In further detail, patients with cancer reported slightly more negative ATOA than individuals without cancer. Hierarchical linear multiple regression found that ATOA contributed significantly to the prediction of physical and mental health after controlling for socio-demographic variables and resilience.

Mikkelsen et al. [141], by means of a semi-structured interview, explored attitudes towards physical activity in a sample of 23 older patients with advanced cancer during palliative oncological treatment. Physical activity was perceived as a positive self-management strategy that allows to obtain physical benefits and prevent social isolation. It has been described as a pause from thoughts and concerns able to give feelings of normality. This positive attitude toward exercise was reinforced by the information received from the physicians.

Realistic attitudes about the process, side effects, and benefits of chemotherapy were found to promote the decision to use this approach in older breast cancer women [142].

Motivation has an impact on a patient’s ability to cope with cancer and adopt health-promoting behaviors and is associated with better outcomes [135]. A study conducted in a sample of 660 older women with early-stage breast cancer found that lower motivation scores were associated with a lower hazard of all-cause mortality over 10 years of follow-up (0.78 at 5 years, 95% confidence intervals 0.52, 1.19, and 0.77 at 10 years, 95% confidence interval 0.59, 1.00) [143]. These associations diminished after adjusting for age, cancer stages, and HRQOL. These results suggested that the association between motivation and cancer mortality is confounded not only by clinical factors but also by the subjective impact of the disease.

Currently, cancer treatments include multimodal treatments (radiotherapy, surgery, chemotherapy)that all contribute to a higher proportion of survivors and increased hope in older adult populations as well. However, as previously reported by Roth AJ et al. [144], for old-age patients, the word “cancer” may mean hopelessness, pain, fear, death, and stigma. The communication of the diagnosis in older adults is often associated with increased psychological distress that may overcome their individual psychological resilience.

The old-age person often experiences a diagnosis of cancer as supersized to his/her multimorbidity, and existing clinical vulnerability and or frailty that may reflect the degree of previous adjustment versus medical illnesses. The patient’s perception may result in increased concerns about disability, dependency, immobility, loss of cognitive competence, disruption of social relationships, and loneliness, as well as pain and discomfort.

##### Psychosocial Dimension: Social Support

Social support has been defined as a network of family, friends, neighbors, and community members that is available in times of need to provide psychological, physical, and financial help to patients with cancer [145]. Such a network plays a major role in improving the ability to cope with the disease, decreasing the stress related to diagnosis and treatment of cancer, and improving clinical and patient-reported outcomes [136,137,146,147]. In older patients, the increase of support needs related to the frailty goes along with the social network’s reduction over the person’s lifespan: combined, these two factors may impact cancer management and outcomes.

Studies have demonstrated that social support in older patients with cancer is related to patient-reported outcomes and psychological state. In a sample of 1280 older women with non-metastatic breast cancer, an association between tangible social support and decline in HRQOL was noted [148]. A cross-sectional study of 1457 older patients with cancer showed that, along with symptom severity and comorbidity, support need is a primary contributor that is negatively associated with the mental component of health status [149]. Moreover, social support may also affect the tolerance of older adults with cancer to cancer therapy: in a prospective study of 500 older adults starting a new line of chemotherapy, poor social support was found to be predictive of an increased risk of non-haematologic toxicity [150].

## 4. Discussion

Current evidence in older adults with cancer mainly relies on the effectiveness and local damage from cancer and its therapies. However, due to the higher rate of cancer survivors, the identification of late-systemic symptoms that frequently occurred years out of the completion of cancer therapies requires further investigation. This set of late occurring symptoms that may include fatigue, pain, gastrointestinal symptoms, or general discomfort and may have a vast impact on long-term function, frailty, quality of life, and patients’ independence.

Burchfield et al. [151] also recently underscored that a variety of neuropsychiatric disorders may form a part of late symptoms, with a common underlying pathophysiology. Indeed, if the inflammatory insult is persistent or exuberant over the acute phase of cancer, higher brain neuroinflammation may develop with impaired central nervous system plasticity that may result in long-term changes in neural pathways maintaining depression, anxiety, and a set of neuropsychiatric disturbances. In particular, a prevalent cluster of neuropsychiatric disorders in neck cancer survivors was observed that included a lack of interest in people, decreased motivation, and tension with irritability, heavily hampering the overall quality of life.

Over the spectrum of cancer, the repetitive environmental stressors related to cancer and its treatments may perturb homeostasis, favoring the progression of frailty. It has been ascertained that older adults with cancer are more likely to become frailer because of both cancer treatments and the cancer itself, both of which play a role as stressors, decreasing functional reserve over time.

Notwithstanding the missing standard definition for frailty and its measurement in clinical practice, defining a patient’s frailty, beyond chronological age, is of key relevance in the early recognition of the biological state of patients who are prone to higher toxicities and unfavorable cancer-related treatment outcomes and overall poorer prognosis. 

Whatever Fried clinical phenotype or the accumulation deficit model of Rockwood is used, the identification of frailty may unveil clinical vulnerabilities in older cancer patients and guide geriatric interventions that are aimed at increasing resilience, maintaining quality of life, and delivering appropriate supportive care.

Moreover, the intertwined role of cancer and frailty is further complicated by the development of neuropsychiatric disorders that may themselves be moderators of frailty expression, as illustrated in Figure 3.

In particular, the neurobehavioral axis of pre-dementia (mild cognitive impairment—MCI) and the transition to dementia is generally sustained by the degrees and progression of frailty and cancer, and their related therapies are known accelerators of neurocognitive impairment in survivorship and late-life.

Further, depression generally co-occurs in the course of cancer disease, and, namely, the inner sense of loneliness, hopelessness, and demoralization may drive weakness, fatigue, hypomobility, and functional deficit along with anorexia, weight loss, reduced mobility, and social vulnerability succumbing faster frailty trajectories.

In addition, the presence of anxiety may supersize patients’ psychological resilience, and the type of cancer and its advanced stage, along with pre-existing multimorbidity or disability, are increasingly recognized as strong risk factors for suicidal risk. It could be hypothesized that this brand-new psychogeriatric construct could drive poorer clinical outcomes in the survivorship of old-age patients with cancer, although further evidence is warranted.

Similarly, sleep disorders such as hypersomnia, insomnia, and circadian disruption may frequently occur in presence of multimorbidity, dementia, delirium, and depression, driving higher functional decline. Sleep disturbances might be a paradigm of perturbed homeostasis that foster diminished functional reserve over time. Additionally, sleep disorders may be interlinked to the pathogenesis of cancer, increasing the bidirectional interplay between sleep–wake cycle, cancer, and frailty.

Poor attitude, a lack of motivation, the subjective patient’s perception after receiving a diagnosis of cancer and the assessment of goals, such as cultural and religious attitudes and how the cancer diagnosis impacts on late-life developmental tasks, are recognized as strong determinants of the effectiveness of cancer-related treatment, overall health status, and survival. Tightly intertwined with the individual’s attitude is the dimension of social support that is increasingly recognized as playing a role in predicting completion of care. Social vulnerability is also associated with frailty because the interplay between social support, the presence of emotionally supportive and qualitative valuable relationship may count for the social tightness that makes the patients more resistant to the treatment and goals, predicting better tolerance to treatments, psychological resilience, and enhanced quality of life.

Frailty reversibility in older adults with cancer is largely unexplored and could be supported by early geriatric interventions that may shape trajectories of survivorship, portending better clinical outcomes and overall quality of care. The delivery of interventions should be targeted to delirium prevention, supporting cognitive decline with cognitive behavioral strategies, counselling, and medical therapy in case of depression, anxiety, or sleep disturbances when appropriate. Similarly, psychotherapy, physical activities, and enhanced social support, home nursing, or community groups have been shown to positively impact on overall compliance to treatment and outcomes over time. These non-oncological management approaches may shape the individual frailty trajectories, reducing the late-life burden of frailty, and the development of geriatric syndromes [152].

Frailty reversibility in older adults with cancer is largely unexplored and could be supported by early geriatric interventions that may counteract, in a preventative way, the accumulation of clinical deficits, mitigating the trajectory of frailty. To enable these tailored geriatric interventions, it should be clearly assessed which dimensional aspects of frailty more accurately portend unfavorable clinical outcome and the delivery of geriatric-based interventions should be the mainstay strategies to prevent and reverse frailty. Namely, supporting cognitive decline with cognitive behavioral strategies, and counselling and the appropriate prescription of medical therapies in case of depression, anxiety, or sleep disturbances should be first line interventions in older adults newly diagnosed with cancer. Similarly, the clinical management of multimorbidity especially in presence of pain or malnutrition along with the periodical drug’s reconciliation could drive higher degree of frailty reversibility throughout the cancer spectrum. In particular, psychotherapy, physical activities, and enhanced social support, home nursing, or community groups have been shown to positively impact on overall compliance to cancer treatment and outcomes over time. The implementation of these geriatric-based clinical management in oncological routine practice may shape the individual frailty trajectories, reducing the late-life burden of frailty and the development of geriatric syndromes [152]. A better understanding of both neuropsychiatric and psychological correlates in older adults with cancer holds the promise to add knowledge in the clinical progression of old-age cancer patients throughout their course of their disease. Neuropsychiatric disorders may occur during the course of cancer and may portend more severe clinical outcomes, accelerating the clinical trajectory of frailty.

## 5. Conclusions

Older adults with cancer need to be embedded in a multidimensional framework, and frailty can be considered to be the backbone for single-disease development and progression. Receiving a diagnosis of cancer in late-life is often supersized to the multimorbidity and disability burden and the pre-existing burden of frailty of old-age patients. Tumor characteristics, treatments, and options are important in oncology, but frailty needs to be considered as a common ground for decision-making.

Currently, the identification of the whole burden of cancer within the frailty status of the single individual is generally dismissed. The immediate effect of cancer treatment and short-term outcomes in old-age patients are unreliable measures of the effectiveness of care. The spectrum of cancer care in older adults increasingly embraces the survival phase, and the continuum of care develops through a multidisciplinary supportive care strategy. In this same context, the impact of neuropsychiatric disorders, given their intrinsic role in the development of frailty, seems to be of key relevance, particularly in the prediction of late-life symptoms.

Eventually, although preliminary in nature, the present findings add more knowledge regarding the multi-component nature of older adults with cancer. Further clinical challenges may lie in the early identification and clinical longitudinal management of neuropsychiatric disorders and psychological individual’s attitude and motivation as mediators of frailty in older adults with cancer, beyond dementia, in order to provide targeted new interventions to deliver better models of care for these highly vulnerable patients. 

## Figures and Tables

**Figure 1 cancers-14-00258-f001:**
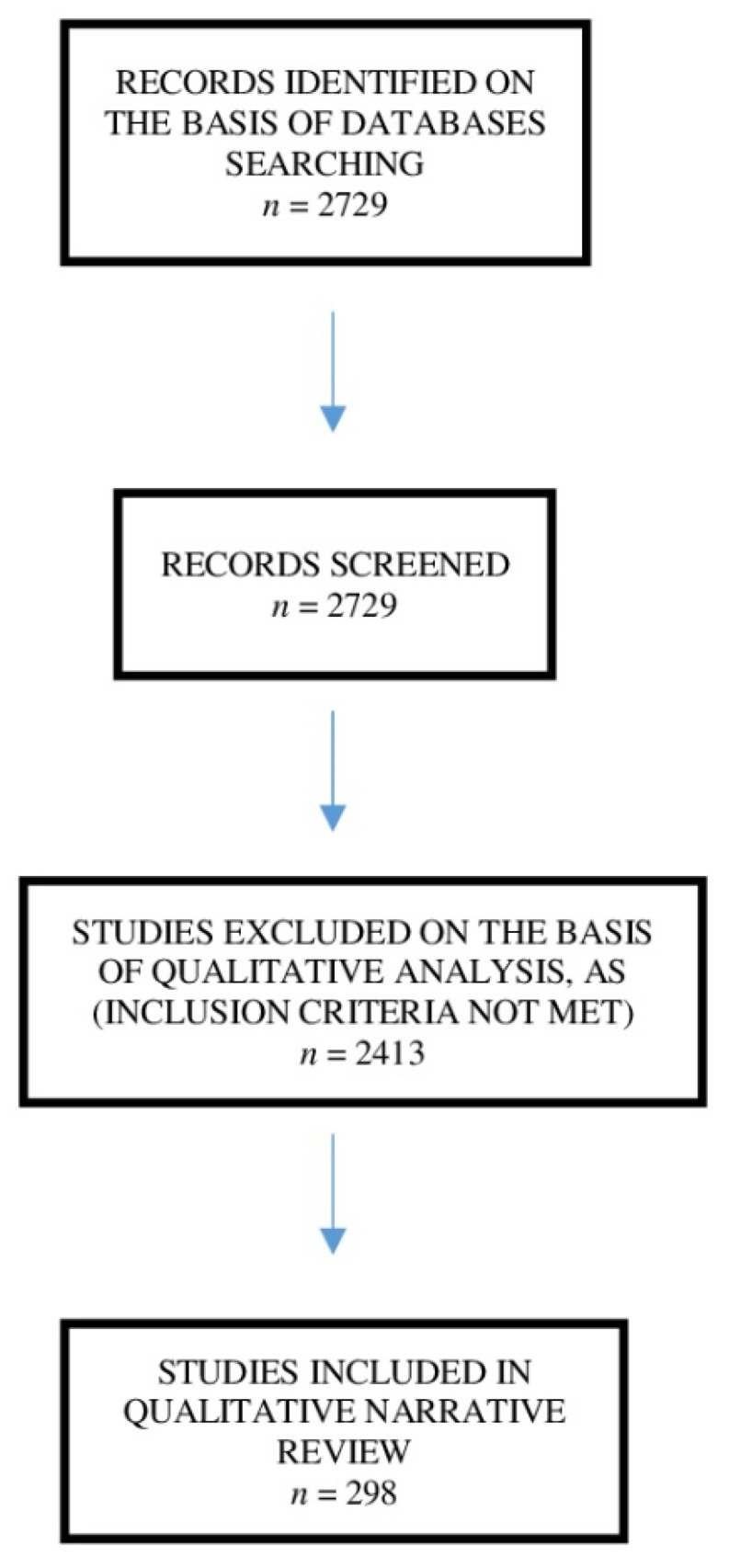
Flowchart diagram of studies selection process according to the inclusion criteria.

**Figure 2 cancers-14-00258-f002:**
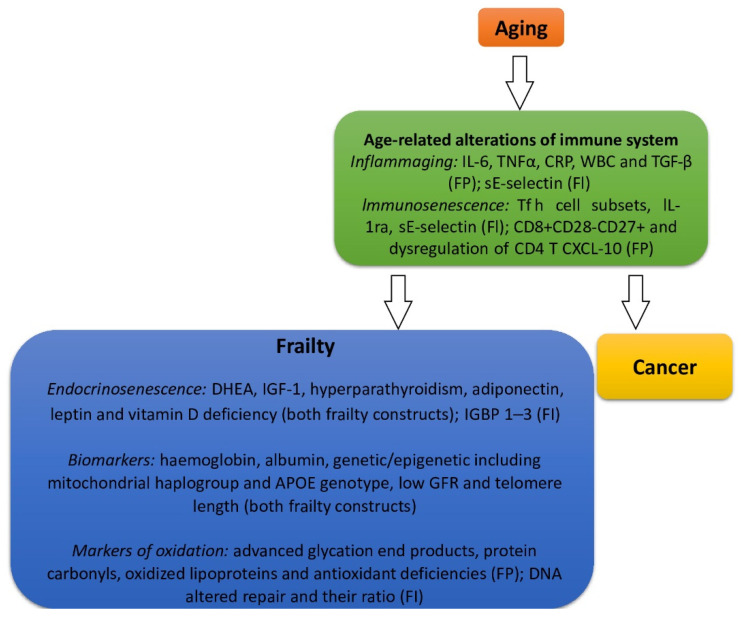
Bidirectional interplay among biological aging, the moderating role of inflammaging and immunosenescence, and frailty biomarkers on the basis of the conceptual frailty framework of the physical frailty phenotype of Fried (FP) or the deficit accumulation model of Rockwood (FI). Interleukin 6 (IL-6), tumor necrosis factor-α (TNF-α), C-reactive protein (CRP) and white blood cells (WBCs), T follicular helper cell (Tfh cell) subsets, interleukin-1 receptor antagonist (IL-1Ra), soluble endothelial leukocyte adhesion molecule-1 (sE-selectin), C-X-C motif chemokine ligand 10 (CXCL10), transforming growth factoR-β (TGF-β), dehydroepiandrosterone (DHEA), insulin-like growth factor-1 (IGF-1), insulin-like growth factor binding protein1–3 (IGFBP 1–3), and glomerular filtration rate (GFR).

**Figure 3 cancers-14-00258-f003:**
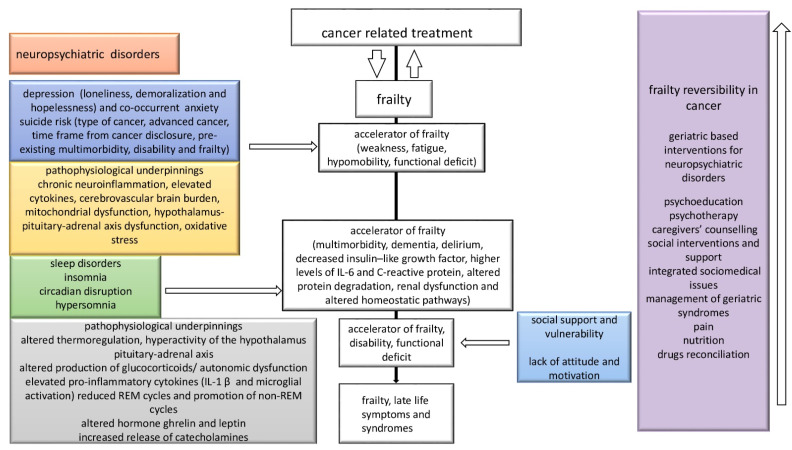
Multidirectional interplay between cancer, frailty, and neuropsychiatric disorders in older adults with cancer and putative progression of frailty and late-life symptoms along with potential frailty reversibility targeted interventions. REM: rapid eye movement.

**Table 1 cancers-14-00258-t001:** Summary of core studies on frailty in older people with cancer and related clinical outcomes on the basis of the two main frailty constructs (i.e., Fried and Rockwood phenotypes).

Reference	Cancer Type/Surgery	Frailty Assessment	Outcome
Pamukcuoglu et al. [18]	HCT	Physical frailty phenotype	Short term: high-grade non-haematological toxicity
			Long term: long-term mortality (1 year)
Kristjansson et al. [21]	Colorectal cancer/elective surgery	Physical frailty phenotype CGA	Short term: postoperative complications and mortality
	-	-	Long term: mortality (median follow-up 20 months)
Tan et al. [20]	Colorectal cancer/elective surgery	Physical frailty phenotype	Short term: postoperative major complications
Burg et al. [22]	Bladder cancer/radical cystectomy	Physical frailty phenotype	Short term: high-grade 30- and 90-day complications
Cespedes Feliciano et al. [23]	Different types of cancer	Physical frailty phenotype	Long term: mortality after cancer diagnosis (median follow-up 5.8 years)
Runzer-Colmenares et al. [24]	Different types of cancer treated with radiotherapy	Physical frailty phenotype, VES-13 ∗, G-8 ∗	Short term: Radiotherapy toxicity of grade III, IV, or V
Runzer-Colmenares et al. [25]	Different types of cancer treated with chemotherapy	Physical frailty phenotype, VES-13 ∗, G-8 ∗	Short term: Chemotherapy toxicity of grade III, IV, or V
Hay et al. [26]	Gynaecologic cancer	Physical frailty phenotype	Short term: administration and completion of chemotherapy
Murillo et al. [19]	Multiple myeloma	Physical frailty phenotype	Long term: mortality (median follow-up 10.6 months)
Inci et al. [28]	Ovarian cancer/cytoreductive surgery	The deficit accumulation model of Rockwood	Short term: severe postoperative complications
			Long term: overall survival ( median follow-up 37.6 months)
Giannotti et al. [29]	Gastrointestinal cancer/surgery	The deficit accumulation model of Rockwood and CGA	Short term: postoperative complications; 30-day mortality
			Long term: 1-year mortality

Haematopoietic Cell Transplant (HCT), Comprehensive Geriatric Assessment (CGA),Vulnerable Elders Survey-13 (VES-13) screening test for frailty, and G-8 questionnaire screening test for frailty. ∗ Both VES-13 and G-8 are based on a two-step screening approach that makes elders eligible for full frailty stratification on the basis of both Fried and/or Rockwood phenotypes.

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
