# Peer review of "Neuropsychiatric Disorders and Frailty in Older Adults over the Spectrum of Cancer: A Narrative Review"

_cancers, 2022, doi:10.3390/cancers14010258_

Round 1

Reviewer 1 Report

The authors have adequately addressed the comments

Author Response

we thank the reviewer for the comment

Reviewer 2 Report

Please find appended my review of this revised manuscript. Although it is substantially improved, it still needs further revision. I am uncertain of the claim that conciseness of the submission is improved. There are still editing issues to be addressed.

Author Response

We thank the reviewer for the raised criticisms. Please find attached   the track version of the revised manuscript and find below the point to point response :

Reviewer's report:

- awkwardly worded sentence that needs to be rewritten ( line 53-57)

Reply: the sentence was rewritten to improve reader's comprehension 

-This is a too long sentence . Please rewrite and divide into shorter sentences (line 72-77)

Reply: the sentence was divided in order to improve reader's comprehension 

-Spelling consistency-e.g. you use both haematological and hematological in the text.Please use the spelling style consistently as provided by the journal guidelines.This issue probably applies elsewhere in the manuscript as well.

Reply: the spelling consistency for ''haematological ''was checked along with other potential spelling inconsistencies within the text 

-Figure needs to be renumbered ( line 332) this is now figure 2 since a new Figure 1 was inserted earlier in the manuscript

Reply: Figures were consecutively renumbered within the text 

-Figure 2 needs to be renumbered ( line 619)

Reply: Figures were consecutively renumbered within the text 

This manuscript is a resubmission of an earlier submission. The following is a list of the peer review reports and author responses from that submission.

Round 1

Reviewer 1 Report

 The association of cancer with frailty and neurocognitive disorders is extremely important given the increasing association of cancer and aging. The authors did a very thourough literature research that may be very helpful to the oncologist and geriatric readers when presesented in a different format.

  Major comments: 

  1. the goals of the review are not specified. I imagine they should include the prevalence and incidence of frailty and neurocognitive disorders in older individuals with cancer and the questions whether these conditions may affect the outcome and maybe a consequence of cancer and cancer treatment
  2. The definition of frailty is missing. The authors need to decide whether they want to adopt just the Fried and Rockwood definitions, in which case they should exclude all other studies or include all studies mentioning frailty. This would probably be a better choice \, but in any case they should describe in a table the definition they use and also which frailty framework is associated with increased risk of complicatios, survival reduction .   Only thanks to this information the reader can decide which definition of fraily may be useful to his/her practice.
  3. The diagnostic criteria of dementia and other neurocognitive disorders should be specified.
  4. 4. Though the grammatics and the spelling are correct, the style is redundant, ripetitive, rambling and generic and does not help the reader.

I make the following recommendations:

  1. Consider to have two diffrent articles ( associaiton with frailty and with neurocognitive disorders). One may also consider three different articles (one dedicated to sleeping disorders and depression)
  2. add the goals of treatment, definition of frailty, diagnostic criteria for neurocognitive diseases
  3. Reduce the text by at least half of the current length, avoiding a lot of hypothesis, concentrating on established facts and using a style more coincise. In particular the current introduction might be eliminated and just substituted with the goals of the study

Reviewer 2 Report

  1. In the Introduction, I failed to understand what previous review to the year 2000 was used as a basis to characterize the current paper as an update.  Please provide a citation that helps the reader understand how the interactions among cancer, neuropsychiatric disorders and frailty have changed approach to patient management as a result of your literature review and manuscript submission.
  2. In section 2, you indicate the search strategy to find relevant scientific publications that might address these issues in older adults, but you don't provide the number of publications identified at first and the net number of papers you were left for you to use as a result of your exclusion criteria.  This information is essential to help the reader understand how broadly your review might apply to this cohort.
  3. Although you indicate that sample sizes <50 were excluded, it is not clear what the cumulative sample size was that supported your interpretation of the implications of these co-morbidities in the management of elderly cancer patients.
  4. There are minor spelling and grammatical issues that need to be eliminated from a further version of this manuscript.